# Urban Courier Delivery in a Smart City: The User Learning Process of Travel Costs Enhanced by Emerging Technologies

**Francesco Russo [1] and Antonio Comi [2],***

1   Dipartimento di Ingegneria dell'Informazione, Delle Infrastrutture e dell'Energia Sostenibile, Mediterranea University of Reggio Calabria, Feo di Vito, 89100 Reggio Calabria, Italy; francesco.russo@unirc.it
2   Department of Enterprise Engineering, University of Rome Tor Vergata, Via del Politecnico 1, 00133 Rome, Italy
*   Correspondence: comi@ing.uniroma2.it

**Abstract:** This paper surveys urban courier routing, pointing out the learning process of the generalized travel cost enhanced by using innovations related to the introduction of emerging information and communication technologies (ICTs, i.e., the internet of things, big data, block chain and artificial intelligence), considering a smart city. Couriers, when planning in advance or choosing the routes in real time for delivering to citizens as well as to business users (including retailers), need to consider both the driving and walking routes (i.e., from the delivery bay to the customers) to optimize their activities. A two-layer literature optimization model is recalled, and the main scientific people-centered challenges that need to be addressed under the light of emerging ICTs are identified and explored, which are the learning process of routing attributes, as well as the opportunity to book on-street delivery bays in advance or in real time. Then, after a literature review on modeling courier activities, a unitary formulation is presented that combines old and real-time network data. In addition, integration with new telematics solutions (i.e., delivery bay booking) is pointed out. Finally, discussions on innovations and cost optimization are presented.

**Keywords:** emerging ICT; city logistics; courier routes; smart city; user-centered solutions; delivery bay booking; last-mile delivering; urban delivery





## 1. Introduction

By creating visions and strategies for urban goods mobility at the regional or local level, several public territorial administrations have begun to understand and address the concerns related to urban goods mobility issues. However, as the COVID-19 pandemic has shown [1], adequate methods for the last-mile delivery of commodities at the inner-city level are frequently lacking [2–8]. In addition, the challenges associated with municipal sustainability are getting worse as the online market continues to expand rapidly [9–11].

Companies must create new distribution models that excel in a variety of areas, including cost-effectiveness, customer satisfaction, and sustainability, to meet the challenges of urban last-mile distribution [12–14]. In this situation, businesses/companies engaged in last-mile distribution, including parcel delivery services, are coming up with a range of approaches. The type and location of logistic facilities, the size of the delivery trucks utilized, the availability of alternative delivery and product exchange sites, and the other factors all affect how the current techniques are different from each other. These particular kind of last-mile distribution methods must take into account a number of aspects in the local context, such as local customers' preferences and demand characteristics, the existing rules, or the operational characteristics of the environment in which they operate.

Currently, when delivering products, couriers frequently encounter traffic jams and delays entering on-street loading/unloading zones (delivery bays) [15], which also has a growing impact on traffic. In addition, truck drivers often lack knowledge and experience about the availability of loading and unloading zones, and in inner-city locations with

many one-way streets and narrow streets, couriers and carriers sometimes make inefficient deliveries [13,16,17]. Distribution productivity is affected by the amount of time spent driving, unloading, and walking [14,18], as well as the length of the routes. In fact, it can be challenging for couriers to find appropriate places to unload the cargo when delivering goods to customers in central business districts (CBD). Searches for the availability of an on-street delivery bay can add a significant amount of time to delivery routes and exacerbate traffic congestion.

Delivery bays can now be reserved in advance or in real time with a dynamic queue thanks to recent advances in sensor technologies, such as presence detectors and the global positioning system (GPS) [16,19]. The best delivery zones to use to minimize the operational and external distribution costs must be known.

The advent of smart cities is changing the use of cities. The term smart city is used in many technical–scientific fields. Each field adopts its own language, but a unified definition that includes all the aspects of a smart city is missing [20,21]. On this note, reference is made to the subsequent specifications given by the European Union, with its different organizations [22–25]. The first reference is from 2010, with the European Union strategy for growth and jobs (Europe 2020 strategy; [22]). The strategy is divided into three growth priorities:

- *smart growth* in the form of efficient spending on innovation, research, and education;
- *sustainable growth*, with regard to the efforts to move towards a low-carbon economy;
- *inclusive growth*, with respect to the initiatives to eradicate poverty and create jobs.

In the context of smart growth, the flagship initiative is innovation, which must be developed with two large groups of players:

- European technology platforms (ETPs), whose stakeholders are industries with the aim of defining research in the medium and long terms by identifying the objectives (*industry driven*);
- European innovation partnerships (EIPs), which represent a new approach to the EU for research and innovation, where public and private entities meet.

To foster sustainability, the European Innovation Partnership for Smart Cities and Communities (EIPSCC; [25]) pushes cities, industries, and citizens to collaborate. Therefore, moving from the concept of smart growth (taking into account innovation, telematic platforms, and thematic forums), the areas of interest refer to: energy production and use (EPU), transport and mobility (TRM), and information and communication technologies (ICTs). Subsequently, the partnership among these sectors is devoted to addressing the progress to improving the services offered by reducing the energy use and consumption of the resources (Figure 1; [23]).

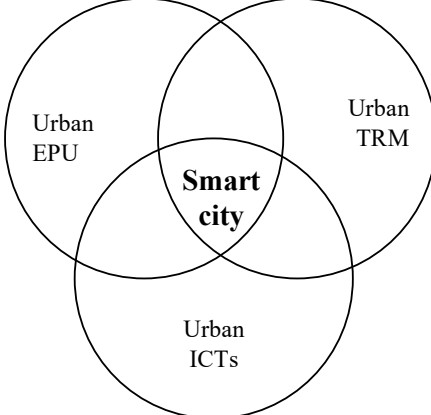

**Figure 1.** The EIP-SCC focus areas [23] for a smart city.

From what has been briefly recalled, it emerged that smart cities are structurally ready to use what is provided by technical–scientific innovations. In this context, this

paper focuses on the models used to identify the optimal courier routes that minimize the delivery (or distribution) costs, considering both the vehicle and walking (i.e., from the delivery bays to the customers) costs.

The paper is structured as follows. Section 2 overviews the emerging technologies in the sphere of city logistics. Section 3 recalls the courier urban routing problem, while Section 4 formalizes the proposed model, evidencing the contribution of different emerging ICTs respect to the different model components using a literature courier modelling framework. Conclusions and further developments are drawn in Section 5.

## 2. Actors of the Emerging Technologies

This work allows one to unify the advances in transport systems and emerging ICTs, having the energy problem as a priority goal. It aims to reduce the generalized cost of low- or zero-emission systems (environmental impact) during use in the final parts of deliveries. The current developments and future directions of people-centered solutions applied to couriers are explored [26,27]. In particular, the analysis evolves to incorporate the learning process of the generalized travel costs related to the innovations deriving from the use of emerging ICTs (e-ICTs) and to point out the benefits both in terms of the internal (operational) and external costs. Therefore, how e-ICTs can contribute to the innovation of courier delivery is investigated and formalized.

E-ICTs are expanding and gaining popularity on a daily basis [28–32] and can be introduced into the study of city logistics. At the current stage of development, it is possible to recognize the four main kinds of technologies that work to advance smart city logistics: internet of things (*IoT*), block chain (*BC*); big data *(BD)*; and artificial intelligence (*AI*). Automation and other technologies that impact vehicle characteristics independently from city logistics and time, e.g., electronic ones for safety or those used to check alcohol levels or whether the driver is sleeping, as mentioned earlier, are not pointed out. For more details, refer to [20,33].

In the context of the courier problem, the emerging technologies impact both the path choice [34,35] to move toward the intermediate customers (e.g., retailers or end consumers) and the delivery bay choice to serve the customers. In fact, referring to the last-mile deliveries, the main stages consist of:

- operations performed in warehouses, where the ICTs can support the management, inventory, and storage activities;
- transport from warehouses to delivery bays and from delivery bays to customers, where the ICTs can support the planning or the routes in advances as well as any delivery decisions that can be taken in real time;
- customer-related tasks, where the ICTs can support the operations performed to deliver to customers (e.g., money transactions and integrity checks).

In particular, *IoT* and *BD* can be considered the main emerging technologies impacting the choice of path between two intermediate stops/deliveries/customers (updating both the path utility and choice model), even though *AI* could better exploit the opportunity offered by real-time information. On the other hand, *BC* allows one to manage the exchanges of values and protected/reserved data of the delivery (in this way, it is also called the internet of values—*IoV*), while AI supports the path choice decisions.

The advantages of city logistics for each user or business (private and/or public) actor, in terms of enhancing their value using the e-ICTs, constitute an important aspect in this context. The following categories of city logistics actors can be regarded as homogeneous with regard to the usage of the above technologies: transport and logistics operators (specifically, transport *enterprises*), *public administrators*, *retailers*, and *end consumers*. Regarding the advantages of courier urban routing, in particular, the key effects/impacts include:

- transport *enterprises*, which, among others, seek to reduce the delivery generalized travel [36–38] costs of at-customer deliveries and reverse logistics; they can employ actualized and/or real-time information from the transportation systems thanks to *IoT*

and *BD*, as described below. In addition, the *BC* allows one to manage the exchanges of values as well as the exchange of business and of performed delivery data;

- *public* territorial *administrations*, which aim to promote the city's sustainable development at all times, making better use of urban public spaces (both for parking and for driving) in relation to all the various demand components (such as passengers and freight using various mode services). They can also provide information on parking availability and the path to follow in real time;
- *retailers*, which might streamline their restocking procedure and include freight-receiving activities in their selling activity, taking the payments into account. They might reduce their estate expenses, minimize (or eliminate) their inventory costs, and take into account the dynamic actualization of the loading space availability to further optimize their role in reverse logistics;
- *end consumers*, which can be citizens who benefit from the decrease in traffic due to city logistics optimizations and the increase in living standards resulting from the increase in safety and reduction of pollution emissions. From another perspective, they are consumers who benefit from future instant deliveries; in fact, they can have real-time information about home deliveries.

## 3. Computerized Vehicle Routing and Scheduling

### 3.1. Problem Definition

When delivering items to clients in central business districts (CBDs), couriers frequently struggle to find appropriate parking places. As mentioned, finding suitable on-street loading zones can take a lot of extra time and have a bigger impact on traffic in terms of congestion, accidents, accessibility, and pollution emissions. The e-ICTs can support couriers to book delivery bays [16,17], considering that the more frequently territorial public administrations introduce time windows that are specific to different classes of vehicles [39], and then they need to organize a dynamic queue. Therefore, there is a need to determine the best delivery bay, which will allow one to minimize the delivery route (times) considering the dynamic slots open and considering that delivery bays are frequently used by couriers to serve many clients at once (Figure 2). As a result, both walking tours and truck cruises have to be included in the optimization of CBD distribution routes [40].

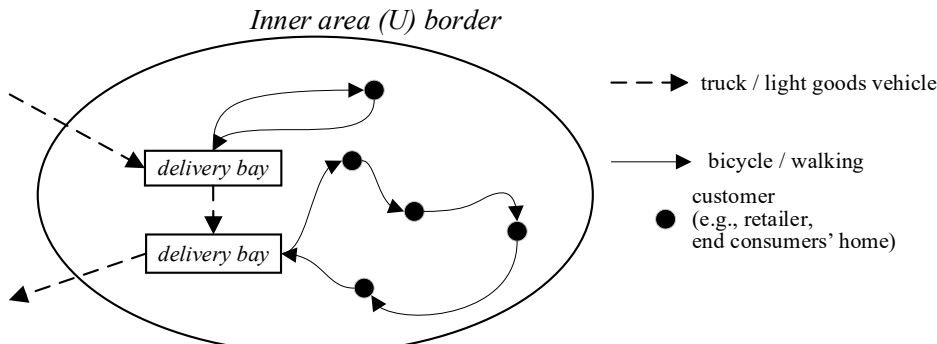

**Figure 2.** Distribution pattern.

In accordance with one of the more recent courier models (i.e., Thompson and Zhang [40]), the problem can be stated and resolved as a multi-objective optimization model by combining the selection and ranking of the delivery zones to be used, as well as the pedestrian routes to move cargo from the delivery bay to the customer (both the shop and the end user's home). The generalized travel costs (such as the kilometers traveled) incurred by walking and driving, along with their respective weights, make up the objective function. Additionally, the limitations (constraints) on the length of time that parking may be allowed are included.

### 3.2. Thompson and Zhang's Formulation

The problem of minimizing the total driving and walking times (cost) can be formulated [40], as follows:

$$\min_{D,W} \alpha \sum_{i,j \in B \cup E} x_{ij} \cdot C_{ij} + (1 - \alpha) \sum_{b \in B} \sum_{i,j \in \{b\} \cup S} z'_{b,i,j} \cdot C'_{ij} \tag{1}$$

This is subject to the following constraints (for flow conservation as well as for topology integrity, and truck and walking route integration).

*Vehicle (Driving) Problem*

$$\sum_{i \in B \cup E} x_{ij} = \sum_{k \in B \cup E} x_{jk} \quad \forall j \in B \cup E \tag{2}$$

This guarantees the conservation of flow at the nodes:

$$\sum_{j \in B \cup E} x_{ij} \leq 1 \quad \forall i \in B \cup E \tag{3}$$

This guarantees that only one path leads from each delivery bay *b* or entrance/exit point *e*:

$$\sum_{j \in E} \sum_{i \in B \cup E} x_{ij} = 1 \tag{4}$$

This guarantees a distinct entrance/exit point; the driving path is closed when it is combined with Equation (2):

$$u_i - u_j + (|B| + 1) \cdot x_{ij} \leq |B| \quad \forall i,j \in B, i \neq j \tag{5}$$

This is used to cut short detours (sub-tours) on a long-distance driving route $|B|$ +1;

$$x_{ij} \in \{0, 1\} \quad \forall i, j \in B \cup E \tag{6}$$

*Walking Problem*

$$\sum_{l \in B} \sum_{i \in B \cup S} z'_{b,is} = 1 \quad \forall s \in S \tag{7}$$

This guarantees a single visit for each consumer:

$$\sum_{k \in B \cup E} x_{kl} \geq z'_{ij} \quad \forall i, j \in \{b\} \cup S, \forall b \in B \tag{8}$$

This shows that each customer is attended to during the walking route, while the delivery bay is visited along the driving route:

$$\sum_{j \in \{b\} \cup S} z'_{b,ij} \leq 1 \quad \forall b \in B \tag{9}$$

This implies that just one route leads to or from each delivery bay:

$$\sum_{j \in \{b\} \cup S} z'_{b,kl} \geq z'_{b,ij} \quad \forall i, j \in \{b\} \cup S, \forall b \in B \tag{10}$$

This guarantees that the pedestrian route from/to the delivery port is completed:

$$\sum_{j \in \{b\} \cup S} z'_{b,ij} = \sum_{j \in \{b\} \cup S} z'_{b,jk} \quad \forall j \in \{b\} \cup S, \forall b \in B \tag{11}$$

This shows that the routes from/to the delivery bays and the clients (customers) are identical:

$$u'_i - u'_j + (|S| + 1) \cdot z'_{b,ij} \leq |S| \quad \forall i, j \in S, i \neq j, \forall b \in B \tag{12}$$

This is used to remove detours from the walking path:

$$z'_{b,ij} \in \{0, 1\} \quad \forall i, j \in B \cup S, \forall b \in B \tag{13}$$

where:

- $\alpha$ is the weight factor when comparing walking and driving;
- $S$ is the collection of customers ($s$), and $B$ is the set of delivery bays ($b$);
- $E$ is the CBD's series of entry/exit points;
- $D$ and $W$ are, respectively, the driving and walking routes;
- $C_{ij}$ is the generalized travel cost from the $i$ to $j$ points via driving along a route belonging to set $D$;
- $C'_{ij}$ is the generalized travel cost from the $i$ to $j$ points via walking along a route belonging to set $W$;
- $x_{ij}$ and $z'_{b,ij}$ binary variables have a value of *1* if the path from $i$ to $j$ is a component of $D$ or $W$ (i.e., to use delivery bay $b$ along the driving route $ij$), respectively, and it is *0* otherwise;
- $u_i$ and $u'_i$ are binary variables equal to *1* if the position of the zone (customer) is $i$ along the driving and walking routes, respectively.

## 4. The Proposed Approach

### 4.1. Dynamic Learning Process

According to the modelling framework recalled earlier and without the loss of generalization if the other courier models in the literature are used, Equation (1) and the linked constraints can be further developed, including the opportunities offered by telematics and, in particular, by the e-ICTs. In fact, the emerging technologies modify the considered approach both for weekly planning and daily dynamics. Given that each network link's data on the previous day can be stored in *BD*, there is an opportunity to use this information in planning in order to forecast the generalized path costs, identify the favored delivery bays, and then determine the best route to go to customers from the selected delivery bay. The usage of *IoT* becomes essential in daily dynamics to update the model after a learning process of the path attributes that rely on the time $\tau$ of day $t$. Therefore, given a generic path for going from location $i$ to location $j$ at time $\tau$ of day $t$, the generalized path costs at time $\tau$ of day $t$, $C[\tau, t]$ are a function of the path attributes, which depend on the time $\tau$ of day $t$, $X[\tau, t]$:

$$C[\tau, t] = \psi(X[\tau, t]) \tag{14}$$

Such attributes can be estimated by the user (in the decision process as a driver and as a walker) according to a process of learning. In general, learning happens simultaneously with the evolution of $\tau$ and $t$. Additionally, the value experienced for various attributes (revealed by all the vehicles on all the network links and stored in *BD*) on the previous days $X[t-1]$, $X[t-2]$ can be considered, while, for the other attributes, the updates that the drivers/walkers know about each time $\tau$ in day $t$ can be pointed out:

$$C[\tau, t] = \psi(X[\tau], X[t-1], X[t-2], \ldots) \tag{15}$$

Therefore, recalling the classical approach to the learning process [36,37], the basic formulation of the attributes of the path for driving from location $i$ to location $j$ is updated using the info coming from the different emerging technologies belonging to the different actors.

Let

- $C_{ij}[\tau, t]$ be the generalized path cost for driving from location $i$ to location $j$ (along a route belonging to the set $D$) at time $\tau$ of day $t$;
- $X_{ij}^{ct^{act}}[\tau, t]$ be the known path cost attributes of each path for travelling from place $i$ to $j$ on time $\tau$ of day $t$, which depend on the used technology, $ct$, to which the class of actors (*act*) belongs. It should be noted that the technologies can be available to public administrations (*PA*; e.g., delivery bay occupancy), transport enterprises (*TLO*; e.g., vehicle sensors), and can also refer to *BC* if, for example, parking payment is performed.;

through the *IoT*, the real-time configuration of the network can be performed; therefore, at the current time $\tau$ of day $\widetilde{t}$, the emerging technology impacts the path attributes as follows:

$$C_{ij}[\tau, \widetilde{t}] = \psi\left( X^{IoT^{PA}}[\tau, \widetilde{t}]; \ X^{IoT^{TLO}}[\tau, \widetilde{t}] \right) \tag{16}$$

- through the *BD*, the prior knowledge on network evolution can be pointed out; therefore, on day $t$, the emerging technology allows the generalized path cost for all the times $\tau$ to be updated, as follows:

$$C_{ij}[\tau, t] = \psi\left( X^{BD,fo}[\tau, t] \right) \tag{17}$$

with $X$ the vector of the path attribute costs, whose $h$-th element represents the $h$-attribute of path from $i$ to $j$ forecasted (*fo*) on day $t$. It is obtained using the equation below:

$$X_{h,ij}^{BD,fo}[t] = \gamma \cdot X_{h,ij}^{BD,\exp}[t-1] + (1 - \gamma) \cdot X_{h,ij}^{BD,fo}[t-1] \tag{18}$$

where

- $X_{h,ij}^{fo}[t]$ is the estimate of attribute $h$-th for the route belonging to the set $D$ forecasted/computed on day $t$;
- $X_{h,ij}^{\exp}[t-1]$ is the estimate of attribute $h$-th for the route belonging to the set $D$ experienced/tested on day $t-1$;
- $\gamma$ ($\in [0, 1]$) is the weight given to the experienced/tested value.

*BC* can then be used to demonstrate legitimacy or authenticity with regard to the direct trip (walking) from the loading zone $b$ to the customer's location and vice versa. Additionally, consumers can gain knowledge about whether a product was sourced ethically, was an original piece, and was preserved in the proper conditions. In addition, the contribution to managing payments is very relevant, including home deliveries. Therefore, Equation (17), specified for the route belonging to set $W$, can be updated as follows:

$$C_{bs}[\tau, t] = \omega\left( X_{bs}, \ X_{bs}^{BC^{PA}}[\tau, t], X_{bs}^{BC^{TLO}}[\tau, t] \right) \tag{19}$$

where $C_{bs}[\tau, t]$ is the generalized travel cost for walking from delivery zone $b$ to customer $s$ on time $\tau$ of day $t$.

The proposed formulation can be beneficial for a large mass of information coming from the network on what has happened on the previous days (e.g., actual network travel times) and on the real-time situation (e.g., real-time travel times). In particular, no specific requirements for the data of the model are different than those that are already available to the operators or to the public administrations [33], where the past and real-time info are combined using an exponential filter through the weight $\xi$.

### 4.2. Preliminary Results

To show the opportunities offered by such an approach, a case study is recalled. It is based on the case study proposed by Russo and Comi [33], where the opportunity to solve delivery issues through path cost updating was explored. This case study refers to deliveries to be performed in Rome and 10 sets of customers that need to be served. In particular, at

each customer location, according to the number of deliveries that need to be performed, a delivery time has been set which includes the time for parking, delivering the items, as well as for finalizing the administrative matters. To avoid overlapping/compensatory effects, only the times have been considered. Therefore, the generalized travel path cost is equal to path travel time (i.e., $X$ = path travel time). The main results are reported in Table 1, where the departure times of the courier from the depot (*DP*) and from each delivery bay (1, ..., 10) are given. An ordered list of customers to be served (i.e., the delivery bays to visit) after each visit is also reported. Finally, the benefit of using the proposed approach is shown in the last column of Table 1 (driving time), where the savings of the real-time calculated solution before they leave the current delivery bay with respect to that of a conventional solution (i.e., the average one) are reported. The driving time is significantly reduced, going over 20% at the end when the truck returns at the depot.

**Table 1.** Sequence of customer visits through average and real-time travel merging ($\xi$ = 0.70) *.

| Departure Time | Order of Delivery Bay to Visit | | | | | | | | | | | | Driving Time (hh:mm:ss) | Δ Driving Time |
|---|---|---|---|---|---|---|---|---|---|---|---|---|---|---|
| *average* | *DP* | *4* | *7* | *8* | *10* | *9* | *2* | *1* | *6* | *3* | *5* | *DP* | *02:28:45* | |
| 09:30 | D | 2 | 1 | 6 | 3 | 5 | 4 | 7 | 9 | 10 | 8 | DP | 02:34:00 | 3.53% |
| 10:15 | 2 | 3 | 1 | 7 | 4 | 6 | 5 | 8 | 9 | 10 | DP | | 02:04:08 | −16.55% |
| 10:53 | 3 | 6 | 1 | 5 | 4 | 7 | 8 | 9 | 10 | DP | | | 01:55:01 | −22.68% |
| 11:17 | 6 | 1 | 5 | 4 | 7 | 8 | 9 | 10 | DP | | | | 02:17:51 | −7.33% |
| 11:38 | 1 | 4 | 5 | 7 | 8 | 9 | 10 | DP | | | | | 02:01:17 | −18.47% |
| 12:03 | 4 | 5 | 7 | 8 | 9 | 10 | DP | | | | | | 02:04:53 | −16.05% |
| ... | | | | | | | | | | | | | | |

Δ = variation with respect to the average sequence of customer visits. *DP* = depot. * source: Russo and Comi [33].

### 4.3. Advanced Courier Routing

The model formulated using the equations of set 1–13 can benefit from the emerging technologies, as explicated by the equations of set 14–19 allows one to update the route/path costs by merging past and real-time info. The capacity to estimate the route costs (disutility/utility; *BD*), determine the best driving and walking paths to the consumers, and identify the set of preferred delivery zones is one of the opportunities provided by e-ICTs in the transportation merging real-time and previous costs to forecast the route costs (*AI*). In addition, the value exchanges can be managed, and more comprehensive track-and-trace capabilities can be provided (*BC*).

In addition, the current changes that are guiding delivery operations in urban areas include:

- *small* and *frequent* shop deliveries (due to a lack of retail storefronts in inner districts, as a result of high rent prices and just-in-time rules), with potential impacts on the availability to have a large supply of shopping services within highly populated areas (spatial proximity);
- *e-commerce* and omni-channel retailing, which are also guided by the evolution of telematics and the new opportunity offered to end consumers thanks to the improvement of digital connectivity;
- *new ways to deliver* products to customers, e.g., express deliveries, same-day deliveries, as well as instant deliveries;
- *reverse logistics*, both for recycling and products that are no longer desired or used.

The modelling framework can be extended in order to include:

- delivering via sustainable means [12,41,42], e.g., bikes;
- delivery at pick-up points, including automated delivery [43–46]. Such systems allow the optimization of deliveries to end consumers, given that they can solve the issues related to failures and sprawling, as well as to have access within the limited traffic zones (LTZ);

- delivery using autonomous delivery robots and drones (ADR; [47–49]). ADRs are motorized vehicles with electric propulsion that can deliver freight or deliveries to customers without the assistance of a delivery worker. The two types of ADRs are road autonomous delivery robots (RADRs), which travel on streets shared with conventional motorized vehicles, and sidewalk autonomous delivery robots (SADRs), which are pedestrian-sized robots that solely use sidewalks or pedestrian routes.

Consequently, e-ICTs became essential for creating sophisticated courier assistance advisors coupled with delivery bay systems [16,50]. It should be noted that after the use of an advanced learning process, the choice updating model can be used to point out the dynamics of the user's choice, upgrading the classical model representing choice behavior into a sequential one [51,52]. In addition, the optimization problem can be expanded to include a further stage for positioning delivery bays within the CBD to lower the operational costs incurred by couriers in accordance with the demand anticipated in the information obtained from *BD* (Figure 3). The findings of the research on two or more echelon problems can be used to improve this formulation [53,54].

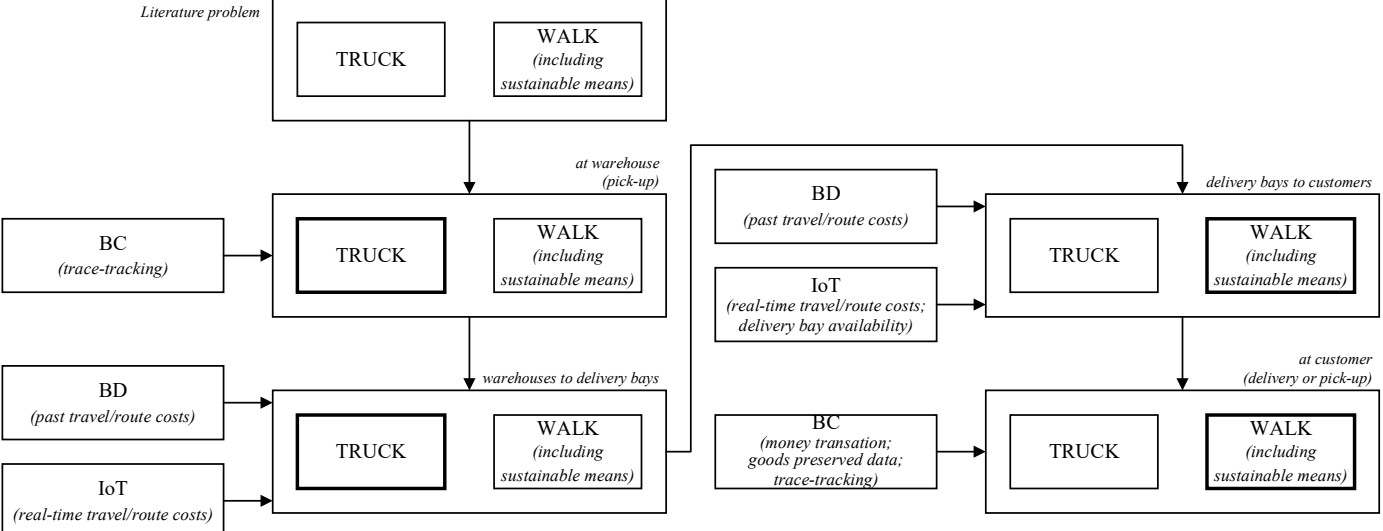

**Figure 3.** Courier routing: the role of e-ICT.

## 5. Discussion and Conclusions

There are many challenges for couriers operating in inner-city areas, in particular, in the context defined by a smart city, where emerging ICTs contribute to solving the real issues of its citizens. The users, in their everyday lives, are faced with some of the most recurring problems that the growth of built-up areas causes, such as impacts from freight transport systems.

The costs of driving and walking must be taken into account while choosing the best routes for the distribution of commodities. This paper highlights the opportunity offered by e-ICTs in pushing advancements in the methods and models used to support courier activities, as well as for reducing the impacts of freight distribution. It does this by evoking the two-layer optimization model in previous literature that generates both driving and walking routes in the context of smart cities and sustainable development. The reference city is the smart one, where the three fields of transportation, ICT, and energy work jointly to increase the quality of the city in line with Agenda 2030 [55]. The innovations pushed by the e-ICTs (internet of things, block chain, big data, and artificial intelligence) are identified in the light of people-centered solutions. In fact, using the emerging technologies to solve the courier routing problem allows one to optimize the costs of driving as well as for walking efficiently using the data ($BD^{PA}$, i.e., owned by public administrations, and $BD^{TLO}$, i.e., owned by enterprises) acquired with sensors ($IoT^{PA}$ and $IoT^{TLO}$). The opportunity to

implement such a multi-objective approach, including energy consumption/reduction, to a location problem for designing nearby delivery areas as well as mobile depots has been focused on, too.

The development of a technology that helps couriers deliver in urban areas using scheduling delivery bays and delivery tour support, as well as providing individualized information to users through real-time data is a key advancement. The tool might enable the users to reserve delivery bays in advance in accordance with the details of their delivery trip in order to carry out delivery operations effectively. This people (user)-centered tool can represent an effective support both for operators and city administrators. As a result, the operators can further cut back on both their operational costs and the time required for delivery activities. Municipalities can lessen the negative effects of urban freight delivery traffic, which will increase the sustainability and livability of the city, when also using some advanced tools for public transport [56,57]. Additionally, future research on the learning process could be conducted, highlighting the en-route tailored information qualities through state preference design and a faster updating procedure. The results of an additional study would also demonstrate the average time and cost savings from its implementation and the ability to extend the suggested framework in order to take advantage of the opportunity given by the digital twin.

**Author Contributions:** Conceptualization, F.R. and A.C.; methodology, F.R. and A.C.; writing—original draft preparation, A.C.; writing—review and editing, F.R. and A.C.; visualization, F.R. and A.C.; supervision, F.R. and A.C. All authors have read and agreed to the published version of the manuscript.

**Funding:** This research received no external funding.

**Institutional Review Board Statement:** Not applicable.

**Informed Consent Statement:** Not applicable.

**Data Availability Statement:** Data are contained within the article. Data are available under request to the authors.

**Acknowledgments:** The authors wish to thank the reviewers for their contribution, which were considerably useful in improving the paper.

**Conflicts of Interest:** The authors declare no conflict of interest.

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
