# Peer review of "Urban Courier Delivery in a Smart City: The User Learning Process of Travel Costs Enhanced by Emerging Technologies"

_sustainability, doi:10.3390/su152316253_

Round 1

Reviewer 1 Report

Comments and Suggestions for Authors

In the manuscript titled “Urban courier delivering in a smart city: the user learning process of travel costs enhanced by emerging technologies”, the authors survey the urban courier routing, pointing out the learning process of travel costs enhanced by innovations related to the introduction of the emerging information and communication technologies. This study contains some interesting findings and are valuable for the understanding of the opportunity offered by e-ICT in pushing advancements in methods and models for supporting courier activities as well as for reducing the impacts of freight distribution. However, even though the methodological approach is explained clearly, there are still several issues that need addressing, as my comments below for details.

Some minor modification suggestions:

(1) Relevant research background needs to be supplemented in INTRODUCTION.

(2) The formatting of references needs to be kept consistent and in line with the requirements of academic journals, with some references missing information such as page numbers;

(3) Some of the formulas are arranged in a slightly confusing way;

(4) Another obvious problem with this paper is lack of sufficient explanation of the simulation results. You need to explain your simulation results in detail and why you got such results;

(5) The public transport, which is an important part of smart city, may include online ride hailing and bikesharing. Just list several as follows.

Understanding operation patterns of urban online ride-hailing services: A case study of Xiamen

Research on carbon emissions of public bikes based on the life cycle theory

Comments on the Quality of English Language

Good

Author Response

In the manuscript titled “Urban courier delivering in a smart city: the user learning process of travel costs enhanced by emerging technologies”, the authors survey the urban courier routing, pointing out the learning process of travel costs enhanced by innovations related to the introduction of the emerging information and communication technologies. This study contains some interesting findings and are valuable for the understanding of the opportunity offered by e-ICT in pushing advancements in methods and models for supporting courier activities as well as for reducing the impacts of freight distribution. However, even though the methodological approach is explained clearly, there are still several issues that need addressing, as my comments below for details.

ANSWER

Thanks so much for the appreciation. We are delighted to revise our paper according to the suggestions provided.

Some minor modification suggestions:

(1) Relevant research background needs to be supplemented in INTRODUCTION.

ANSWER

Done. The following studies were reviewed and included:

  • Zhang, L.; Ding, P.; Thompson, R.G. A Stochastic Formulation of the Two-Echelon Vehicle Routing and Loading Bay Reservation Problem. Transportation Research Part E: Logistics and Transportation Review 2023, 177, 103252, doi:10.1016/j.tre.2023.103252.
  • Dalla Chiara, G.; Krutein, K.F.; Ranjbari, A.; Goodchild, A. Providing Curb Availability Information to Delivery Drivers Reduces Cruising for Parking. Sci Rep 2022, 12, 19355, doi:10.1038/s41598-022-23987-z.
  • Grangier, P.; Gendreau, M.; Lehuédé, F.; Rousseau, L.-M. An Adaptive Large Neighborhood Search for the Two-Echelon Multiple-Trip Vehicle Routing Problem with Satellite Synchronization. European Journal of Operational Research 2016, 254, 80–91, doi:10.1016/j.ejor.2016.03.040.

(2) The formatting of references needs to be kept consistent and in line with the requirements of academic journals, with some references missing information such as page numbers;

ANSWER

We used the automated reference tool, ZOTERO, that allows to edit reference list according to MDPI standard.

(3) Some of the formulas are arranged in a slightly confusing way;

ANSWER

Done.

(4) Another obvious problem with this paper is lack of sufficient explanation of the simulation results. You need to explain your simulation results in detail and why you got such results;

ANSWER

Done.

(5) The public transport, which is an important part of smart city, may include online ride hailing and bikesharing. Just list several as follows.

Understanding operation patterns of urban online ride-hailing services: A case study of Xiamen

Research on carbon emissions of public bikes based on the life cycle theory

ANSWER

Done.

Reviewer 2 Report

Comments and Suggestions for Authors

The article provides a comprehensive overview of the complex challenges and opportunities in the field of urban courier delivery in the context of a smart city. It combines a review of existing literature with a forward-looking perspective on emerging ICT and their potential impact on the courier industry.

The article takes a holistic approach to understanding urban courier delivery, considering both driving and walking routes. This broad perspective is essential in addressing the intricacies of last-mile logistics in a smart city.

The article effectively highlights the role of emerging technologies such as the Internet of Things, big data, blockchain, and artificial intelligence in optimizing travel costs and enhancing the efficiency of courier services. It recognizes the transformative potential of these technologies.

There are a few suggestions that could help improve the content and clarity of the research:

1.               The introduction is too long, could have been shortened a little bit.

2.               It is necessary to correct the design of formulas in the article. Why are formulas aligned on the left edge and not in the middle? The formulae should be after the formulae and not inside the formulae, e.g. formulae (1.8), (1.9).

3.               In 32 sources of literature you refer to your article which has not yet been published, is it in press? If not, it is not correct.

Author Response

The article provides a comprehensive overview of the complex challenges and opportunities in the field of urban courier delivery in the context of a smart city. It combines a review of existing literature with a forward-looking perspective on emerging ICT and their potential impact on the courier industry.

The article takes a holistic approach to understanding urban courier delivery, considering both driving and walking routes. This broad perspective is essential in addressing the intricacies of last-mile logistics in a smart city.

The article effectively highlights the role of emerging technologies such as the Internet of Things, big data, blockchain, and artificial intelligence in optimizing travel costs and enhancing the efficiency of courier services. It recognizes the transformative potential of these technologies.

ANSWER

Thanks so much for the appreciation. We are delighted to revise our paper according to the suggestions provided.

There are a few suggestions that could help improve the content and clarity of the research:

  1. The introduction is too long, could have been shortened a little bit.

ANSWER

Done.

  1. It is necessary to correct the design of formulas in the article. Why are formulas aligned on the left edge and not in the middle? The formulae should be after the formulae and not inside the formulae, e.g. formulae (1.8), (1.9).

ANSWER

Done.

  1. In 32 sources of literature you refer to your article which has not yet been published, is it in press? If not, it is not correct.

ANSWER

Yes, in press.

Reviewer 3 Report

Comments and Suggestions for Authors

This article proposes an algorithm for optimising traveling costs and time for delivering commodities, under the prism of smart city. Although a very promising topic, some improvements should be made, so that this article can have a high impact in its field.

More specifically, a comparison with similar research and suggested methodologies should be made, so as to pinpoint the contribution of the specific model to research and to the improvement of courier traveling in practice. A more thorough discussion on the results of the given example which would clarify the findings of this research and how it contributes to the optimisation of traveling time and costs should be made. Also, the phrase "to avoid overlapping / compensatory effects, only times have been considered" (l. 305-306) is not convincing; costs should also be shown and discussed in the example, as the article's title is dedicated to them.

In addition, a  detailed explanation of the functionality and the assumptions of the proposed algorithm should be given; e.g. what do travel costs include? Energy costs? Salary costs? Third part costs? Any costs, whose boundaries are set by the user?

On the trivial side, in line 218 it is stated that: "D and W are respectively the walking and driving routes" - from my understanding of the paper, it should be: "D and W are respectively the driving and walking routes".

Comments on the Quality of English Language

The article must be proof read, so as to improve English, to make it more understandable. E.g. in line 28 the term "urban goods" must be an awkward translation, referring to "commodities", while there are some syntax errors that may confuse the reader, e.g. the COVID-19 epidemic has out (l. 28-29) / with respect to of efforts (l. 66), etc.

Author Response

This article proposes an algorithm for optimising traveling costs and time for delivering commodities, under the prism of smart city. Although a very promising topic, some improvements should be made, so that this article can have a high impact in its field.

More specifically, a comparison with similar research and suggested methodologies should be made, so as to pinpoint the contribution of the specific model to research and to the improvement of courier traveling in practice. A more thorough discussion on the results of the given example which would clarify the findings of this research and how it contributes to the optimisation of traveling time and costs should be made. Also, the phrase "to avoid overlapping / compensatory effects, only times have been considered" (l. 305-306) is not convincing; costs should also be shown and discussed in the example, as the article's title is dedicated to them.

ANSWER

The first reviewer's note requires some specific responses.

The first answer concerns the “comparison to be carried out with similar research and suggested methodologies, in order to identify the contribution of the specific model to the research and improvement of courier travel practice.” In the introduction, all the most important models present in the literature were recalled both in terms of identifying the routes for the courier and for the choice. The combination of the two models reduces the available literature to a few works. The article introduces the problem of the “learning process” for travel costs. To the authors’ knowledge, there is no work in the literature that deals with this problem together with the two mentioned above, therefore it is the first time that an overall model that introduces learning in a discrete choice, made on path alternatives to be identified.

The second theme introduced concerns “a more in-depth discussion on the results of the example provided which would clarify the findings of this research and how it contributes to the optimization of travel times and costs.” The example was introduced to demonstrate the applicability of the proposed model and the simplicity of implementation in a real case. significant results are reported in the Table 1. The main comment introduced concerns the reduction of time which can exceed peaks of 20%. We did not want to give particular emphasis to the numerical results which are extremely positive, because it could appear as an operation of over-evaluation of the proposed model, therefore we chose to keep the attention focused on the quality of the model and its easy implementability, precisely without emphasizing the high quality of the results. Furthermore, the high quality emerges directly only by seeing the 20% reduction compared to using the average tour.

The third point concerns “avoiding overlaps/compensatory effects only the times were considered”. The problem of the effects of attributes with different trends is found in all cities where city logistics policies are introduced. In these cities, the trend of monetary costs for tolls and parking depending on the timetables defined by the specific policies determine overall values, based on the utility of the individual user, profoundly different starting from the same distances, or, even with identical topologies of the city. Therefore, we are led back to a proxy of utility which is precisely time as highlighted in the next point. The use of the proxy allows us to follow the results of the model without having to consider the policies of the specific cities.

The last point concerns the proposed methodology that uses generalized travel costs (Cascetta, 2011; Ortuzar and Willumsen, 2011, Daganzo; Ghiani Laporte and Musmanno, 2004), expressed as the combination of different route attributes. In the example discussed in the paper, the generalized travel cost is expressed only as a function of the travel time which, in the first instance, can be considered a good proxy of the total “costs” incurred by operators in last mile delivery, avoiding as previously said to use more attributes, thus making it more complex to explain which properties of the model and which are those of the costs (or of the policies).

In addition, a detailed explanation of the functionality and the assumptions of the proposed algorithm should be given; e.g. what do travel costs include? Energy costs? Salary costs? Third part costs? Any costs, whose boundaries are set by the user?

ANSWER

Yes, in general the generalised travel costs can include all the recalled attributes.

On the trivial side, in line 218 it is stated that: "D and W are respectively the walking and driving routes" - from my understanding of the paper, it should be: "D and W are respectively the driving and walking routes".

ANSWER

Done

Comments on the Quality of English Language. The article must be proof read, so as to improve English, to make it more understandable. E.g. in line 28 the term "urban goods" must be an awkward translation, referring to "commodities", while there are some syntax errors that may confuse the reader, e.g. the COVID-19 epidemic has out (l. 28-29) / with respect to of efforts (l. 66), etc.

ANSWER

Done

Reviewer 4 Report

Comments and Suggestions for Authors

This article is of interest because it discusses the current problem of choosing optimal routes for the goods distribution, as well as the opportunities provided by electronic ICT to promote progress in methods and models for supporting courier activities, the use of which will improve the sustainability of the transport and logistics system. The article title adequately reflects the content. In the abstract, the authors provide the article essence, briefly describe the state of the problem, research methods, and results. Keywords correspond to the article content.

In the introduction, the authors provide a brief literature review on the  article topic, indicate the goal and objectives of the study, and also provide the structure of the article. The second section is devoted to a description of the use of ICT and the IoT in distribution and transport. The third part is devoted to a description of computerized routing and vehicle scheduling. In the fourth section, the authors describe the proposed approach: dynamic learning process, preliminary results, and discuss advanced courier routing. In the “Discussion and conclusions” section, the authors summarize the results obtained and provide conclusions on the work.

The article has been prepared in accordance with the instructions for authors and is relevant to the topic it is researching and publishing. In our opinion, the article corresponds to the topic “optimization of courier service routing” and is similar in type to Preliminary Research.

Comment.

1.      The topic is interesting, however, in our opinion, the discussion is superficial: there are several comments on the chosen approach to improving the courier service. Firstly, if the goal is to reduce transportation in urban areas, then it would be more logical to use distributed delivery: a vehicle brings the cargo to a pickup point, from where it is delivered by courier to a customer located in the nearest area, and customers can also pick up the order themselves. Secondly, the work does not say anything about the convenience of the customer, which is, in principle, incorrect, since a long wait due to optimization of delivery from the courier’s point of view can lead to long waits and negate the benefits of courier delivery.

2.      Taking into account the previous remark, in our opinion, it would be good if the authors first listed all the problems that may arise during courier delivery, all types and delivery options (by type of cargo and its characteristics - dimensions, special storage conditions, etc.), otherwise it turns out that the author actually considered only routing, although he used multi-criteria analysis.

3.      In general, it is unclear how a literature review with a relevant and attractive topic reflects the scientific and practical components of existing research, as well as how the proposed method will help improve the performance of the courier service in practice.

4.      It is necessary to improve the quality of figure 3: increase the font size or change its type, since small inscriptions are difficult to read.

Author Response

This article is of interest because it discusses the current problem of choosing optimal routes for the goods distribution, as well as the opportunities provided by electronic ICT to promote progress in methods and models for supporting courier activities, the use of which will improve the sustainability of the transport and logistics system. The article title adequately reflects the content. In the abstract, the authors provide the article essence, briefly describe the state of the problem, research methods, and results. Keywords correspond to the article content.

In the introduction, the authors provide a brief literature review on the article topic, indicate the goal and objectives of the study, and also provide the structure of the article. The second section is devoted to a description of the use of ICT and the IoT in distribution and transport. The third part is devoted to a description of computerized routing and vehicle scheduling. In the fourth section, the authors describe the proposed approach: dynamic learning process, preliminary results, and discuss advanced courier routing. In the “Discussion and conclusions” section, the authors summarize the results obtained and provide conclusions on the work.

The article has been prepared in accordance with the instructions for authors and is relevant to the topic it is researching and publishing. In our opinion, the article corresponds to the topic “optimization of courier service routing” and is similar in type to Preliminary Research.

Comment.

  1. The topic is interesting, however, in our opinion, the discussion is superficial: there are several comments on the chosen approach to improving the courier service. Firstly, if the goal is to reduce transportation in urban areas, then it would be more logical to use distributed delivery: a vehicle brings the cargo to a pickup point, from where it is delivered by courier to a customer located in the nearest area, and customers can also pick up the order themselves.

ANSWER

Our study is placed in the context of urban realities under given conditions, on which the courier cannot intervene, that is the general situation for courier. A city may or may not be equipped with a network of pickup points of various types and with various logics. The problem remains the same: how do we improve a courier route that needs to serve more points? The problem studied in the paper is not that of pickup point networks. Furthermore, the observation made by the auditor according to which it is “more logical” to use another model for the last mile is not fully acceptable. Non-acceptance emerges both from the scientific literature which presents different results for various methods of organizing pickup point networks, and from real processes where it has been observed several times that the consumer “constrained” to use pick up points can review his/her own choice and return to traditional purchasing, leaving from Amazon or other. Naturally, as shown in the literature and recalled in the document (section 4.3), the use of a network of collection points could represent a valid solution for delivery if integrated with the possibility left to the consumer to decide whether to use the pickup points or the home delivery. Therefore, our study continues to be very relevant to optimize all scenarios, with any density of the pickup point network.

Secondly, the work does not say anything about the convenience of the customer, which is, in principle, incorrect, since a long wait due to optimization of delivery from the courier’s point of view can lead to long waits and negate the benefits of courier delivery.

ANSWER

The model highlights the operators' activities and the optimization of delivery routes does not affect customers since they continue to be served according to any constraints (for example, time windows for receiving products) that customers give. The work is considering the utility of the courier and therefore the discrete choice of the courier. It is not a behavioural model of the consumer who must choose based on the service supplied.

If the delivery must be made with deterministic and fixed times, the problem must be posed differently and constraints must be placed on the times by introducing penalty functions for early and late. It is possible to make this extension, but the fundamental core of the paper moves on to a different topic from the one addressed here. The hypothesis in this paper is that for the consumer the “instant” delivery time is indifferent, as long as the delivery takes place within a well-defined time slot. If the attention evolves towards the instant time, it is necessary to use mathematical models for the simulation of the offer including diachronic networks and so on.

  1. Taking into account the previous remark, in our opinion, it would be good if the authors first listed all the problems that may arise during courier delivery, all types and delivery options (by type of cargo and its characteristics - dimensions, special storage conditions, etc.), otherwise it turns out that the author actually considered only routing, although he used multi-criteria analysis.

ANSWER

As mentioned before, the proposed model arises from the advancement of currently used models which provide for the calculation of the best tour at given times with a deterministic structure. The article introduces the possibility that the tour calculation is improved by the information that can reach the general user from ICT devices. All other elements of the problem remain constant. “the type of load and its characteristics” does not influence the possibility that the courier improves the tour by having news (on real time from ICT) that congested conditions have been created on an link and therefore with times higher than the average ones. Similarly, the size of the delivery does not prevent the courier from taking less time. Evidently if the dimensions are such as to require a very specific path, the improvement from ICT information will be zero. The same evaluations must be made for “special storage conditions”.

  1. In general, it is unclear how a literature review with a relevant and attractive topic reflects the scientific and practical components of existing research, as well as how the proposed method will help improve the performance of the courier service in practice.

ANSWER

It is shown in the results of the case study reported in the Table 1. The couriers can significantly reduce their operations costs continuing to respect the needs (and constraints) of their customers. In the introduction, all the most important models present in the literature were recalled both in terms of identifying the routes for the courier and for the choice. The combination of the two models reduces the available literature to a few works. The article introduces the problem of the “learning process” for travel costs. To the authors’ knowledge, there is no work in the literature that deals with this problem together with the two mentioned above, therefore it is the first time that an overall model that introduces learning in a discrete choice, made on path alternatives to be identified.

  1. It is necessary to improve the quality of figure 3: increase the font size or change its type, since small inscriptions are difficult to read.

ANSWER

Done

General conclusion.

The article is devoted to a current topic, since the problems of choosing optimal routes for the distribution of goods, as well as the opportunities provided by electronic ICTs to promote progress in methods and models for supporting courier activities, the use of which will increase the sustainability of the transport and logistics system, and significantly affect the population life quality. In our opinion, the article can be published after modifications and responses to relevant comments. In general, the article is certainly progressive in scientific and practical terms and deserves serious attention from scientists, practitioners and managers in the organization of transport processes and logistics.

ANSWER

Thanks so much for the appreciation. We are delighted to revise our paper according to the suggestions provided.

Reviewer 5 Report

Comments and Suggestions for Authors

The article has a very good introduction and the methodology used is congruent with the proposed objectives (which, although not explicitly presented, can easily be grasped). Apart from the fact that the text opens up more widely the scientific avenues already outlined, its importance lies above all in the fact that it demonstrates the possibility of linking research with the public interest, considered simultaneously on several functional scales: political, economic and individual.

Author Response

The article has a very good introduction and the methodology used is congruent with the proposed objectives (which, although not explicitly presented, can easily be grasped). Apart from the fact that the text opens up more widely the scientific avenues already outlined, its importance lies above all in the fact that it demonstrates the possibility of linking research with the public interest, considered simultaneously on several functional scales: political, economic and individual.

ANSWER

Thanks so much for the appreciation.

Round 2

Reviewer 1 Report

Comments and Suggestions for Authors

The authors have dealt with all my concerns.

Reviewer 3 Report

Comments and Suggestions for Authors

The article has improved, in relation to its former version.